# Reservoir Computing meets Recurrent Kernels and Structured Transforms

**Jonathan Dong**[*1,2]    **Ruben Ohana**[*1,3]    **Mushegh Rafayelyan**[2]    **Florent Krzakala**[1,3,4]

[1]Laboratoire de Physique de l'Ecole Normale Supérieure, ENS, Université PSL, CNRS,
Sorbonne Université, Université de Paris, F-75005 Paris, France
[2]Laboratoire Kastler Brossel, Ecole Normale Supérieure, Université PSL, CNRS,
Sorbonne Université, Collège de France, F-75005 Paris, France
[3]LightOn, F-75002 Paris, France
[4]IdePHICS lab, Ecole Polytechnique Fédérale de Lausanne, Switzerland

## Abstract

Reservoir Computing is a class of simple yet efficient Recurrent Neural Networks where internal weights are fixed at random and only a linear output layer is trained. In the large size limit, such random neural networks have a deep connection with kernel methods. Our contributions are threefold: a) We rigorously establish the recurrent kernel limit of Reservoir Computing and prove its convergence. b) We test our models on chaotic time series prediction, a classic but challenging benchmark in Reservoir Computing, and show how the Recurrent Kernel is competitive and computationally efficient when the number of data points remains moderate. c) When the number of samples is too large, we leverage the success of structured Random Features for kernel approximation by introducing Structured Reservoir Computing. The two proposed methods, Recurrent Kernel and Structured Reservoir Computing, turn out to be much faster and more memory-efficient than conventional Reservoir Computing.

## 1 Introduction

Understanding Neural networks in general, and how to train Recurrent Neural Networks (RNNs) in particular, remains a central question in modern machine learning. Indeed, backpropagation in recurrent architectures faces the problem of exploding or vanishing gradients [1, 2], reducing the effectiveness of gradient-based optimization algorithms. While there exist very powerful and complex RNNs for modern machine learning tasks, interesting questions still remain regarding simpler ones. In particular, Reservoir Computing (RC) is a class of simple but efficient Recurrent Neural Networks introduced in [3] with the Echo-State Network, where internal weights are fixed randomly and only a last linear layer is trained [4]. As the training reduces to a well-understood linear regression, Reservoir Computing enables us to investigate separately the complexity of neuron activations in RNNs. With a few hyperparameters, we can tune the dynamics of the reservoir from stable to chaotic and performances are increased when RC operates close to the chaotic regime [5].

Despite its simplicity, Reservoir Computing is not fully efficient: computational and memory costs grow quadratically with the number of neurons. To tackle this issue, efficient computation schemes have been proposed based on sparse weight matrices [5]. Moreover, there is an active community developing novel hardware solutions for energy-efficient, low-latency RC [6]. Based on dedicated electronics [7–10], optical computing [11–15], or other original physical designs [16], they leverage the robustness and flexibility of RC. Reservoir Computing has already been used in a variety of tasks, such as speech recognition and robotics [17] but also combined with Random Convolutional Neural Networks for image recognition [18] and Reinforcement Learning [19]. A very promising

application today is chaotic time series prediction, where the RC dynamics close to chaos may prove a very important asset [6]. Reservoir Computing also represents an important model in computational neuroscience, as parallels can be drawn with specific regions of the brain behaving like a set of randomly-connected neurons [20].

As RC embeds input data in a high-dimensional reservoir, it has already been linked with kernel methods [5], but merely as an interesting interpretation for discussion. In our opinion, this point of view has not been exploited to its full potential yet. A study derived the explicit formula of the corresponding recurrent kernel associated with RC [21], this important result meaning the infinite-width limit of RC is a deterministic Recurrent Kernel (RK). Still, no theoretical study of convergence towards this limit has been conducted previously and the computational complexity of Recurrent Kernels has not been derived yet.

In this work, we draw the link between RC and the rich literature on Random Features for kernel approximation [22–26]. To accelerate and scale-up the computation of Random Features, one can use optical implementations [27, 28] or structured transforms [29, 30], providing a very efficient method for kernel approximation. Structured transforms such as the Fourier or Hadamard transforms can be computed in $O(n \log n)$ complexity and, coupled with random diagonal matrices, they can replace the dense random matrix used in Random Features.

Finally, we note that Reservoir Computing can be unrolled through time and interpreted as a multilayer perceptron. The theoretical study of such randomized neural networks through the lens of kernel methods has attracted a lot of attention recently [31–33], which provides a further motivation to our work. Some parallels were already drawn between Recurrent Neural Networks and kernel methods [34, 35], but they do not tackle the high-dimensional random case of Reservoir Computing.

**Main contributions —** Our goal in this paper is to bridge the gap between the considerable amount of results on kernels methods, random features — structured or not — and Reservoir Computing.

First, we rigorously prove the convergence of Reservoir Computing towards Recurrent Kernels provided standard assumptions and derive convergence rates in $O(1/\sqrt{N})$, with $N$ being the number of neurons. We then numerically show convergence is achieved in a large variety of cases and does not occur in practice only when the activation function is unbounded (for instance with ReLU).

When the number of training points is large, the complexity of RK grows; this is a common drawback of kernel methods. To circumvent this issue, we propose to accelerate conventional Reservoir Computing by replacing the dense random weight matrix with a structured transform. In practice, Structured Reservoir Computing (SRC) allows to scale to very large reservoir sizes easily, as it is faster and more memory-efficient than conventional Reservoir Computing, without compromising performance.

These techniques are tested on chaotic time series prediction, and they all present comparable results in the large-dimensional setting. We also derive the computational complexities of each algorithm and detail how Recurrent Kernels can be implemented efficiently. In the end, the two acceleration techniques we propose are faster than Reservoir Computing and can tackle equally complex tasks. A public repository is available at `https://github.com/rubenohana/Reservoir-computing-kernels`.

## 2   Recurrent Kernels and Structured Reservoir Computing

Here, we briefly describe the main concepts used in this paper. We recall the definition of Reservoir Computing and Random Features, define Recurrent Kernels (RKs) and introduce Structured Reservoir Computing (SRC).

**Reservoir Computing (RC)** as a Recurrent Neural Network receives a sequential input $i^{(t)} \in \mathbb{R}^d$, for $t \in \mathbb{N}$. We denote by $x^{(t)} \in \mathbb{R}^N$ the state of the reservoir, $N$ being the number of neurons in the reservoir. Its dynamics is given by the following recurrent equation:

$$x^{(t+1)} = \frac{1}{\sqrt{N}} f\left(W_r \, x^{(t)} + W_i \, i^{(t)}\right) \tag{1}$$

where $W_r \in \mathbb{R}^{N \times N}$ and $W_i \in \mathbb{R}^{N \times d}$ are respectively the reservoir and input weight matrices. They are fixed and random: each weight is drawn according to an i.i.d. gaussian distribution with variances

$\sigma_r^2$ and $\sigma_i^2$, respectively. Finally, $f$ is an element-wise non-linearity, typically a hyperbolic tangent. To refine the control of the reservoir dynamics, it is possible to add a random bias and a leak rate. In the following, we will keep the minimal formalism of Eq. (1) for conciseness.

We use the reservoir to learn how to predict a given output $o^{(t)} \in \mathbb{R}^c$ for example. The output predicted by the network $\hat{o}^{(t)}$ is obtained after a final layer:

$$\hat{o}^{(t)} = W_o \, x^{(t)} \tag{2}$$

Since only these output weights $W_o \in \mathbb{R}^{c \times N}$ are trained, the optimization problem boils down to linear regression. Training is typically not a limiting factor in RC, in sharp contrast with other neural network architectures. The expressivity and power of Reservoir Computing rather lies in the high-dimensional non-linear dynamics of the reservoir.

**Kernel methods** are non-parametric approaches to learning. Essentially, a kernel is a function measuring a correlation between two points $u, v \in \mathbb{R}^p$. A specificity of kernels is that they can be expressed as the inner product of feature maps $\varphi : \mathbb{R}^p \to \mathcal{H}$ in a possibly infinite-dimensional Hilbert space $\mathcal{H}$, i.e. $k(u, v) = \langle \varphi(u), \varphi(v) \rangle_{\mathcal{H}}$. Kernel methods enable the use of linear methods in the non-linear feature space $\mathcal{H}$. Famous examples of kernel functions are the Gaussian kernel $k(u, v) = \exp\left(-\frac{\|u - v\|^2}{2\sigma^2}\right)$ or the arcsine kernel $k(u, v) = \frac{2}{\pi} \arcsin \frac{\langle u, v \rangle}{\|u\| \|v\|}$. When the dataset becomes large, it is expensive to numerically compute the kernels between all pairs of data points.

**Random Features** have been developed in [22] to overcome this issue. This celebrated technique introduces a random mapping $\phi : \mathbb{R}^p \to \mathbb{R}^N$ such that the kernel is approximated in expectation:

$$k(u, v) = \langle \varphi(u), \varphi(v) \rangle_{\mathcal{H}} \approx \langle \phi(u), \phi(v) \rangle_{\mathbb{R}^N} \tag{3}$$

with $\phi(u) = \frac{1}{\sqrt{N}}[f(\langle w_1, u \rangle), ..., f(\langle w_N, u \rangle)]^\top \in \mathbb{R}^N$ and random vectors $w_1, ..., w_N \in \mathbb{R}^p$. Depending on $f$ and the distribution of $\{w_i\}_{i=1}^N$, we can approximate different kernel functions.

There are two major classes of kernel functions: translation-invariant **(TI)** kernels and rotation-invariant **(RI)** kernels. In our study, we will consider TI kernels of the form $k(u, v) = k(\|u - v\|_2^2)$ and RI kernels of the form $k(u, v) = k(\langle u, v \rangle)$. Both can be approximated using Random Features [22, 36]. For example, Random Fourier Features (RFFs) defined by:

$$\phi(u) = \frac{1}{\sqrt{N}}[\cos(\langle w_1, u \rangle), ..., \cos(\langle w_N, u \rangle), \sin(\langle w_1, u \rangle), ..., \sin(\langle w_N, u \rangle)]^\top \tag{4}$$

approximate any TI kernel (provided $k(0) = 1$). For example, when $w_1, ..., w_N \sim \mathcal{N}(0, \sigma^{-2} I_p)$, we approximate the Gaussian kernel. A detailed taxonomy of Random Features can be found in [37].

Random Features can be more computationally efficient than kernel methods, when their number $N$ is smaller than the number of data points $n$. For this particular reason, Random Features are a method of choice to implement large-scale kernel-based methods.

**Link with Reservoir Computing.** It is straightforward to notice that reservoir iterations of Eq. (1) can be interpreted as a Random Feature embedding of a vector $[x^{(t)}, i^{(t)}]$ (of dimension $p = N + d$), multiplied by $W = [W_r, W_i]$. This means the inner product between two reservoirs $x^{(t)}, y^{(t)}$ driven respectively by two inputs $i^{(t)}$ and $j^{(t)}$ converges to a deterministic kernel as $N$ tends to infinity:

$$\langle x^{(t+1)}, y^{(t+1)} \rangle \approx k([x^{(t)}, i^{(t)}], [y^{(t)}, j^{(t)}]) \tag{5}$$

As explained previously, this kernel depends on the choice of $f$ and the distribution of $W_r$ and $W_i$.

By denoting $l^{(t)} = \sigma_i^2 \langle i^{(t)}, j^{(t)} \rangle$ and $\Delta^{(t)} = \sigma_i^2 \|i^{(t)} - j^{(t)}\|^2$, TI and RI kernels are then of the form:

$$k([x^{(t)}, i^{(t)}], [y^{(t)}, j^{(t)}]) = k(\sigma_r^2 \langle x^{(t)}, y^{(t)} \rangle + l^{(t)}) \quad \text{(RI)} \tag{6}$$
$$= k(\sigma_r^2 \|x^{(t)} - y^{(t)}\|^2 + \Delta^{(t)}) \quad \text{(TI)} \tag{7}$$

**The Recurrent Kernel limit.** Looking at Eq. (6) and (7), we notice the kernel at time $t$ depends on approximations of kernels at previous times in a recursive manner. Here, we introduce Recurrent Kernels to remove the dependence in $x^{(t)}$ and $y^{(t)}$.

We suppose for the sake of simplicity $x^{(0)} = y^{(0)} = 0$. We define RI recurrent kernels as:

$$\begin{cases} k_1\left(l^{(0)}\right) = k\left(l^{(0)}\right) \\ k_{t+1}\left(l^{(t)}, ..., l^{(0)}\right) = k\left(\sigma_r^2 k_t\left(l^{(t-1)}, ..., l^{(0)}\right) + l^{(t)}\right), \quad \text{for } t \in \mathbb{N}^* \end{cases} \tag{8}$$

Similarly for TI recurrent kernels with Random Fourier Features, exploiting the property in Eq. (4) that $\|x^{(t)}\|^2 = \|y^{(t)}\|^2 = 1$:

$$\begin{cases} k_1\left(\Delta^{(0)}\right) = k\left(\Delta^{(0)}\right) \\ k_{t+1}\left(\Delta^{(t)}, ..., \Delta^{(0)}\right) = k\left(\sigma_r^2\left(2 - 2k_t\left(\Delta^{(t-1)}, ..., \Delta^{(0)}\right)\right) + \Delta^{(t)}\right), \quad \text{for } t \in \mathbb{N}^* \end{cases} \tag{9}$$

These Recurrent Kernel definitions describe hypothetical asymptotic limits of large-dimensional Reservoir Computing, interpreted as recurrent Random Features. We will study in Section 3.1 the convergence towards this limit.

**Structured Reservoir Computing.** In the Random Features literature, it is common to use structured transforms to speed-up computations of random matrix multiplications [29, 30]. They have also been introduced for trained architectures, with Deep [38] and Recurrent Neural Networks [39].

We propose to replace the dense random weight matrices $W = [W_r, W_i]$ by a succession of Hadamard matrices $H$ (structured orthonormal matrices composed of $\pm 1/\sqrt{p}$ components) and diagonal random matrices $D_i$ for $i = 1, 2, 3$ sampled from an i.i.d. Rademacher distribution [30]:

$$W = \frac{\sqrt{p}}{\sigma} H D_1 H D_2 H D_3 \tag{10}$$

We use the Hadamard transform for its simplicity and the availability of high-performance libraries in [40]. This structured transform provides the two main properties of a dense random matrix: mixing the activation of the neurons (Hadamard transform) and randomness (diagonal matrices).

## 3 Convergence theorem and computational complexity

### 3.1 Convergence rates

Our first main result is a convergence theorem of Reservoir Computing to its kernel limit. We use Bernstein's concentration inequality in our recurrent setting. Several assumptions will be necessary:

- The kernel function $k$ is Lipschitz-continuous with constant $L$, i.e. $|k(a) - k(b)| \leq L|a - b|$.
- The random matrices $W_r$ and $W_i$ are resampled for each $t$ to obtain uncorrelated reservoir updates: $x^{(t+1)} = \frac{1}{\sqrt{N}} f(W_r^{(t)} x^{(t)} + W_i^{(t)} i^{(t)})$. This assumption is required for our theoretical proof of convergence, but we show convergence is reached numerically even without redrawing the weight matrices, which is standard in Reservoir Computing (in Fig. 1).
- The function $f$ is bounded by a constant $\kappa$ almost surely, i.e. $|f(W_{res}^{(t)} x^{(t)} + W_{in}^{(t)} i^{(t)})| \leq \kappa$.

**Theorem 1.** *(Rotation-invariant kernels) For the RI recurrent kernel defined in Eq. (8), under the assumptions detailed above, and with $\Lambda = \sigma_r^2 L$. For all $t \in \mathbb{N}$, the following inequality is satisfied for any $\delta > 0$ with probability at least $1 - 2(t + 1)\delta$:*

$$\left| \langle x^{(t+1)}, y^{(t+1)} \rangle - k_{t+1}(l^{(t)}, ..., l^{(0)}) \right| \leq \frac{1 - \Lambda^{t+1}}{1 - \Lambda} \Theta(N) \qquad if \quad \Lambda \neq 1 \tag{11}$$

$$\leq (t+1)\Theta(N) \qquad if \quad \Lambda = 1 \tag{12}$$

*with $\Theta(N) = \frac{4\kappa^2 \log \frac{1}{\delta}}{3N} + 2\kappa^2 \sqrt{\frac{2 \log \frac{1}{\delta}}{N}}$.*

*Proof.* We use the following Proposition (Theorem 3 of [41] restated in Proposition 1 of [24]):

**Proposition 1.** *(Bernstein inequality for a sum of random variables). Let $X_1, ..., X_N$ be a sequence of i.i.d. random variables on $\mathbb{R}$ with zero mean. If there exist $R, S \in \mathbb{R}$ such that $|X_i| \leq R$ almost everywhere and $\mathbb{E}[X_i^2] \leq S$ for $i \in \{1, ..., N\}$, then for any $\delta > 0$ the following holds with probability at least $1 - 2\delta$:*

$$\left| \frac{1}{N} \sum_{i=1}^{N} X_i \right| \leq \frac{2R \log \frac{1}{\delta}}{3N} + \sqrt{\frac{2S \log \frac{1}{\delta}}{N}} \tag{13}$$

Under the assumptions, Proposition 1 yields with probability greater than $1 - 2\delta$:

$$\left|\langle x^{(t+1)}, y^{(t+1)}\rangle - k([x^{(t)}, i^{(t)}], [y^{(t)}, j^{(t)}])\right| \leq \frac{4\kappa^2 \log\frac{1}{\delta}}{3N} + 2\kappa^2\sqrt{\frac{2\log\frac{1}{\delta}}{N}} = \Theta(N) \qquad (14)$$

It means the larger the reservoir, the more Random Features $N$ we sample, and the more the inner product of reservoir states concentrates towards its expectation value, at a rate $O(1/\sqrt{N})$. We now apply this inequality recursively to complete the proof, based on the observation that both Eq. (11) and (12) are equivalent to: $\left|\langle x^{(t+1)}, y^{(t+1)}\rangle - k_{t+1}(l^{(t)}, ..., l^{(0)})\right| \leq (1 + \Lambda + \Lambda^2 + ... + \Lambda^t)\Theta(N)$.

For $t = 0$, provided $x^{(0)} = y^{(0)} = 0$, we have, according to Eq. 14, with probability at least $1 - 2\delta$:

$$\left|\langle x^{(1)}, y^{(1)}\rangle - k_1(l^{(0)})\right| \leq \Theta(N) \qquad (15)$$

For any time $t \in \mathbb{N}^*$, let us assume the following event $A_t$ is true with probability $\mathbb{P}(A_t) \geq 1 - 2t\delta$:

$$\left|\langle x^{(t)}, y^{(t)}\rangle - k_t(l^{(t-1)}, ..., l^{(0)})\right| \leq (1 + ... + \Lambda^{t-1})\Theta(N) \qquad (16)$$

Using the Lipschitz-continuity of $k$, this inequality is equivalent to:

$$\left|k(\sigma_r^2\langle x^{(t)}, y^{(t)}\rangle + l^{(t)}) - k(\sigma_r^2 k_t(l^{(t-1)}, ..., l^{(0)}) + l^{(t)})\right| \leq (\Lambda + ... + \Lambda^t)\Theta(N) \qquad (17)$$

With Eq. (14), the following event $B_t$ is true with probability $\mathbb{P}(B_t) \geq 1 - 2\delta$:

$$\left|\langle x^{(t+1)}, y^{(t+1)}\rangle - k(\sigma_r^2\langle x^{(t)}, y^{(t)}\rangle + l^{(t)})\right| \leq \Theta(N) \qquad (18)$$

Summing Eq. (17) and (18), with the triangular inequality and a union bound, the following event $A_{t+1}$ is true with probability $\mathbb{P}(A_{t+1}) \geq \mathbb{P}(B_t \cap A_t) = \mathbb{P}(B_t) + \mathbb{P}(A_t) - \mathbb{P}(B_t \cup A_t) \geq 1 - 2\delta + 1 - 2t\delta - 1 \geq 1 - 2(t+1)\delta$:

$$\left|\langle x^{(t+1)}, y^{(t+1)}\rangle - k_{t+1}(l^{(t)}, ..., l^{(0)})\right| \leq (1 + ... + \Lambda^t)\Theta(N) \qquad (19)$$

$\square$

A statement and proof of a similar convergence bound for TI recurrent kernels is provided in the Supplementary Material.

### 3.2 Numerical study of convergence

The previous theoretical study required three important assumptions that may not be valid for Reservoir Computing in practice. Moreover, there is still no rigorous proof on the convergence of Structured Random Features in the non-recurrent case due to the difficulty to deal with correlations between them. Thus, we numerically investigate whether convergence of RC and SRC towards the Recurrent Kernel limit is achieved in practice.

In Fig. 1, we numerically compute the Mean-Squared Error (MSE) between the inner products obtained with a Recurrent Kernel and RC/SRC for different number of neurons in the reservoir. We generate 50 i.i.d. gaussian input time series $i_k^{(t)}$ of length $T$, for $k = 1, \ldots, 50$ and $t = 0, \ldots, T-1$. Each time series is fed into 50 reservoirs that share the same random weights, for RC and SRC. We compute the MSE between inner products of pairs of final reservoir states $\langle x_k^{(T)}, x_l^{(T)}\rangle$ and the deterministic limit obtained directly with $k_T(i_k^{(T-1)}, i_l^{(T-1)}, \ldots, i_k^{(0)}, i_l^{(0)})$, for all $k, l = 1, \ldots, 50$. The computation is vectorized to be efficiently implemented on a GPU. Three different activation functions, the rectified linear unit (ReLU), the error function (Erf), and Random Fourier Features defined in Eq. (4), have been tested with different variances of the reservoir weights. The larger the reservoir weights, the more unstable the reservoir dynamics becomes.

Nonetheless, convergence is achieved in a large variety of settings, even when the assumptions of the previous theorem are not satisfied. For example, the ReLU non-linearity is not bounded and

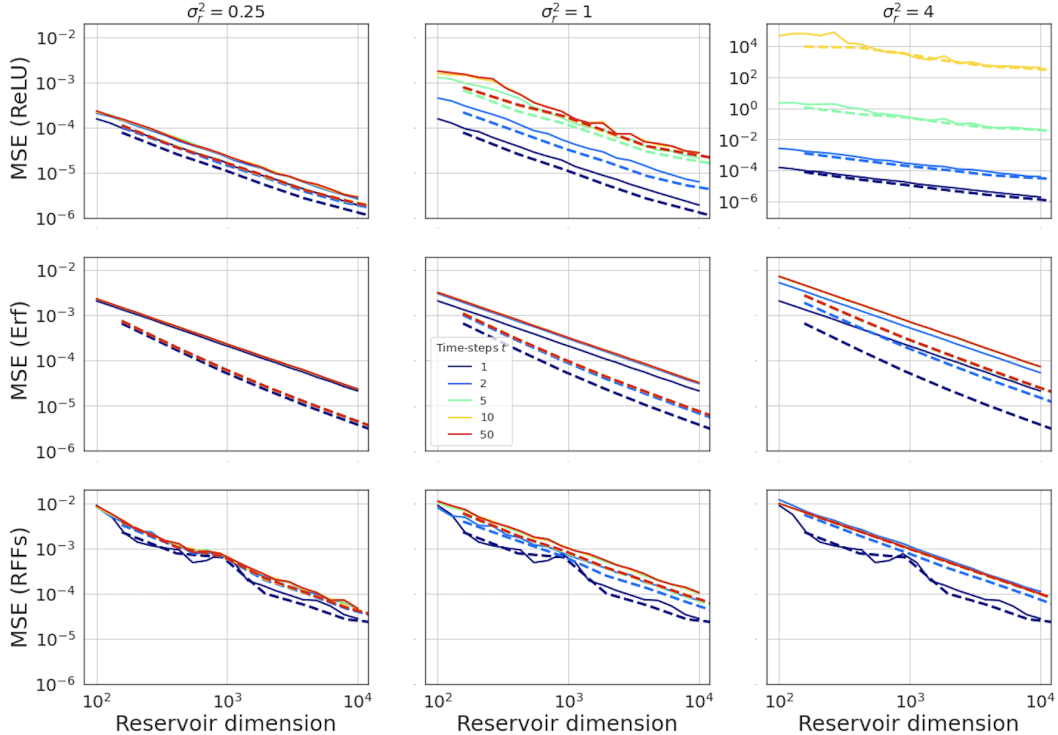

Figure 1: Convergence of Reservoir Computing towards its Recurrent Kernel limit for different variances of the reservoir weights $\sigma_r^2$ (columns), activation functions (lines: ReLU, Erf, RFFs) and times, for RC **(solid lines)** and SRC **(dashed lines)**. We observe that for the two bounded activation functions (Erf and RFFs), RC always converge towards the RK limit even at large times $t$. For ReLU, RC converges when $\sigma_r^2 = 0.25$ and 1, and diverges as $t$ increases when $\sigma_r^2 = 4$. We also observe that SRC always yields equal or faster convergence than RC. The MSE decreases with an $O(1/N)$ scaling, which is consistent with the convergence rates derived in Theorem 1.

converges when $\sigma_r^2 \geq 1$. It is interesting to notice even for a large variance $\sigma_r^2 = 4$ do Reservoir Computing and Structured Reservoir Computing converge towards the RK limit for the second and third activation functions. This behavior has been consistently observed with any bounded $f$.

On the other hand, Structured Reservoir Computing seems to always converge faster than Reservoir Computing. We thus confirm in the recurrent case the intriguing effectiveness of Structured Random Features [42], that may originate from the orthogonality of the matrix $W_r$ in SRC.

As a final remark, weight matrices in Fig. 1 were not redrawn as supposed in Section 3.1. This assumption was necessary as correlations are often difficult to take into account in a theoretical setting. This is important for Reservoir Computing as it would be unrealistically slow to draw new random matrices at each time step.

### 3.3   When to use RK or SRC?

The two proposed alternatives to Reservoir Computing, Recurrent Kernels and Structured Reservoir Computing, are computationally efficient. To understand which algorithm to use for chaotic system prediction, we need to focus on the limiting operation in the whole pipeline of Reservoir Computing, the recurrent iterations. They correspond to Eq. (1) for RC/SRC and Eq. (8, 9) for RK. We have a time series of dimension $d$, that we split into train/test datasets of lengths $n$ and $m$ respectively. The exact computational and memory complexities of each step are described in Table 1.

**Forward:** In both Reservoir Computing and Structured Reservoir Computing, Eq. (1) needs to be repeated as many times as the length of the time series. For Reservoir Computing, it requires a multiplication by a dense $N \times N$ matrix, the associated complexity scales as $O(N^2)$. On the other hand, Structured Reservoir Computing uses a succession of Hadamard and diagonal matrix multiplications, reducing the complexity per iteration to $O(N \log N)$.

| | Reservoir Computing | Structured Reservoir Computing | Recurrent Kernel |
|---|---|---|---|
| Forward | $O(nN^2)$ | $O(nN \log N)$ | $O(n^2\tau)$ |
| Training | $O(nN^2 + N^3)$ | $O(nN^2 + N^3)$ | $O(n^3)$ |
| Prediction | $O(mN^2)$ | $O(mN \log N)$ | $O(mn\tau)$ |
| Memory | $O(nN + N^2)$ | $O(nN)$ | $O(n^2 + mn)$ |

Table 1: Computational and memory complexity of the three algorithms. SRC accelerates the forward pass and decreases memory complexity compared to conventional RC. The complexity of RK depends on the number of training and testing points and would be advantageous when $n \ll N$.

Recurrent Kernels need to recurrently compute Eq. (8, 9) for all pairs of input points. For chaotic time series prediction, this corresponds to a $n \times n$ kernel matrix for training, and another kernel matrix of size $n \times m$ for testing. To keep computation manageable, we use a well-known property in Reservoir Computing, called the Echo-State Property: the reservoir state should not depend on the initialization of the network, i.e. the reservoir needs to have a finite memory $\tau$. This property is important in Reservoir Computing and has been studied extensively [3, 43–45]. Transposed in the Recurrent Kernel setting, it means we can fix the number of iterations of Eq. (8, 9) to $\tau$, by using a sliding window to construct shorter time series if necessary. A preliminary numerical study of the stability of Recurrent Kernels is presented in the Supplementary.

**Training** requires, after a forward pass on the training dataset, to solve an $n \times N$ linear system for RC/SRC and a $n \times n$ linear system for RK. It is important to note SRC and RK do not accelerate this linear training step. We will use Ridge Regression with regularization parameter $\alpha$ to learn $W_o$.

**Prediction** in Reservoir Computing and Structured Reservoir Computing only requires the computation of reservoir states and multiplication by the learned output weights. Recurrent Kernels need to compute a new kernel matrix for every pair $(i_r, j_q)$ with $i_r$ in the training set and $j_q$ in the testing set. Note that the prediction step includes a forward pass on the test set, followed by a linear model.

## 4 Chaotic time series prediction

**Chaotic time series prediction** is a task arising in many different fields such as fluid dynamics, financial or weather forecasts. By definition, it is difficult to predict their future evolution since initially small differences get amplified exponentially. Recurrent Neural Networks and in particular Reservoir Computing represent very powerful tools to solve this task [46, 47].

The Kuramoto-Sivashinsky (KS) chaotic system is defined by a fourth-order partial derivative equation in space and time [48, 49]. We use a discretized version from a publicly available code [47] with input dimension $d = 100$. Time is normalized by the Lyapunov exponent $\lambda = 0.043$ which defines the characteristic time of exponential divergence of a chaotic system, i.e. $|\delta x^{(t)}| \approx e^{\lambda t}|\delta x^{(0)}|$.

---

**Algorithm 1:** Recurrent Kernel algorithm

---

**Result:** Predictions $\hat{o}^{(t)} \in \mathbb{R}^{c \times m}$
**Input:** A train set $\{i_r^{(t)}\}_{r=1}^n \in \mathbb{R}^{\tau \times d}$ with outputs $o \in \mathbb{R}^{c \times n}$, a test set $\{j_q^{(t)}\}_{q=1}^m \in \mathbb{R}^{\tau \times d}$.
**Training:** Initialize an $n \times n$ kernel matrix $G^{(0)} = 0$;
**for** $t = 0, \ldots, \tau - 1$ **do**
　Compute $G_{rs}^{(t+1)} = k_{t+1}(i_r^{(t)}, i_s^{(t)}, \ldots, i_r^{(0)}, i_s^{(0)})$ using Eq. (8) or (9) and $G_{rs}^{(t)}$.
**end**
Compute the output weights $W_o \in \mathbb{R}^{c \times n}$ that minimize $\|o - W_o G^{(\tau)}\|_2^2 + \alpha\|W_o\|_2^2$;
**Prediction:** Initialize an $n \times m$ kernel matrix $K^{(0)} = 0$;
**for** $t = 1, \ldots, \tau$ **do**
　Compute $K_{rq}^{(t+1)} = k_{t+1}(i_r^{(t)}, j_q^{(t)}, \ldots, i_r^{(0)}, j_q^{(0)})$ using Eq. (8) or (9) and $K_{rq}^{(t)}$.
**end**
Compute the predicted outputs $\hat{o}^{(t)} = W_o K^{(\tau)}$;

---

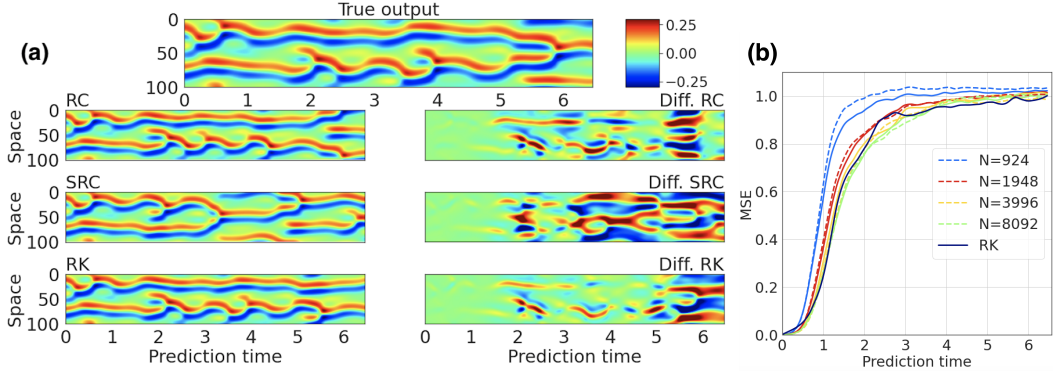

Figure 2: (a) Comparison of different algorithms for the prediction of the Kuramoto-Sivashinsky dataset. True output (top), predictions of RC/SRC/RK (left) and differences with the true output (right), with reservoirs in RC/SRC of size $N = 3{,}996$. We observe that each technique is able to predict up to a few characteristic times. (b) Mean-Squared Error as a function of the prediction time for RC (**full lines**), SRC (**dashed lines**), and RK (**black**). For all the reservoir sizes considered, the performances of RC and SRC are very close and they converge for large dimensions to the RK limit.

KS data points $i^{(0)}, \ldots, i^{(t-1)}$ are fed to the algorithm. The output in Eq. (2) for Reservoir Computing consists here in predicting the next state of the system: $\hat{o}^{(t)} = i^{(t)}$. This prediction is then used for updating the reservoir state in Eq. (1), the algorithm outputs the next prediction $\hat{o}^{(t+1)}$, and we repeat this operation. Thus, Reservoir Computing defines a trained *autonomous dynamical system* that one wants to be synchronized with the chaotic time series [46].

The hyperparameters are found with a grid search, and the same set is used for RC, SRC, and RK to demonstrate their equivalence. To improve the performance of the final algorithm, we also add a random bias and use a concatenation of the reservoir state and the current input for prediction, replacing Eq. (2) by $\hat{o}^t = W_o[x^{(t)}, i^{(t)}]$.

**Prediction performance** is presented in Fig. 2. RC and SRC are trained on $n = 70{,}000$ training points and RK on a sub-sampling of 7,000 of these training points, due to memory constraints. The testing dataset length was set at 2,000. The sizes $N$ in Reservoir Computing and Structured Reservoir Computing are chosen so the dimension $p = N + d$ in Eq. (10) is a power of two for the multiplication by Hadamard matrix. Linear regression is solved using Cholesky decomposition.

The predictions in Fig. 2 show that all three algorithms are able to predict up to a few characteristic times. Since the prediction performance varies quite significantly between different realizations, we also display the Mean-Squared Error (MSE) of each algorithm, as a function of the prediction time and averaged over 10 realizations. We normalize each curve by the MSE between two independent KS systems.

We observe a decrease in the MSE when the size of the reservoir increases, meaning a larger reservoir yields better predictions. Performances are equivalent between RC and SRC, and they converge towards the RK performance for large reservoir sizes. Hence, this means RC, SRC, and RK can seamlessly replace one another in practical applications.

**Timing benchmark.** Several steps in the Reservoir Computing pipeline need to be assessed separately, as described in 3.3. We present the timings on a training set of length $n = 10{,}000$ and testing length of $m = 2{,}000$ in Table 2 for all three algorithms.

The *forward pass*, i.e. computing the recurrent iterations of each algorithm, is considered separately from the linear regression for *training*, to emphasize the cost of this important step. In RC, the most expensive operation is the dense matrix multiplication; the GPU memory was not large enough to store the square weight matrix for the two largest reservoir sizes. With Structured Reservoir Computing, this forward pass becomes very efficient even at large sizes, and memory is not an issue anymore. We observe that the forward pass complexity becomes approximately constant until dimension $\sim 10^5$. On the other hand, Recurrent Kernels iterations are very fast, as we only need to compute element-wise operations in a kernel matrix.

*Prediction* requires a forward pass and then is performed with autonomous dynamics as presented on Fig. 2 where Eq. (2) is repeated 600 times. For Recurrent Kernels, prediction remains slow, and this drawback is exacerbated by the autonomous dynamics strategy in time series prediction, that requires successive prediction steps.

This shows that SRC is a very efficient way to scale-up Reservoir Computing to large sizes and reach the asymptotic limit of performance. On the other hand, the deterministic Recurrent Kernels are surprisingly fast to iterate, at the cost of a relatively slow prediction when the number of training samples $n$ is large.

| | $N = 1,948$ | $N = 3,996$ | $N = 8,092$ | $N = 16,284$ | $N = 32,668$ |
|---|---|---|---|---|---|
| RC | **2.6**/0.02/1.9 | **3.1**/0.05/4.6 | 10.4/0.16/15.4 | Mem. Err. | Mem. Err. |
| SRC | 3.3/0.02/**1.6** | 3.4/0.05/**2.7** | **3.5**/0.16/**3.7** | **3.6**/0.57/**6.8** | **3.6**/2.57/**13.0** |
| RK | | | **0.7**/**0.09**/23.0 | | |

Table 2: Timing (Forward/Train/Predict, in seconds) for a KS prediction task as a function of $N$. We observe that Recurrent Kernels are surprisingly fast, except for prediction. Structured Reservoir Computing reduces drastically the speed of the forward pass at large sizes and is more memory-efficient than Reservoir Computing. Experiments were run on an NVIDIA V100 16GB.

## 5    Conclusion

In this work, we strengthened the connection between Reservoir Computing and kernel methods based on theoretical and numerical results, and showed how efficient implementations of Recurrent Kernels can be competitive with standard RC for chaotic time series prediction. Future lines of work include a deeper study of stability and the extension to different recurrent networks topologies. We deeply think this connection between random RNNs and kernel methods will open up future research on this important topic in machine learning.

We additionally introduced Structured Reservoir Computing, an acceleration technique of Reservoir Computing using fast Hadamard transforms. With only a simple change of the reservoir weights, we are able to speed up and reduce the memory cost of Reservoir Computing and therefore reach very large network sizes. We believe Structured Reservoir Computing offers a promising alternative to conventional Reservoir Computing, replacing it whenever large reservoir sizes are required.

## Broader Impact

Our work consists in a theoretical and numerical study of acceleration techniques for random RNNs. Theoretical studies are important to understand machine learning to avoid relying on black boxes, towards a more responsible use of these algorithms as more and more applications appear in our daily life.

On the other hand, efficient machine learning is necessary due to the ever-increasing power consumption required for computation. The Recurrent Kernels and Structured Reservoir Computing methods we developed pave the way towards much more efficient Reservoir Computing algorithms.

## Acknowledgments and Disclosure of Funding

Authors would like to thank Sylvain Gigan, Antoine Boniface (Laboratoire Kastler-Brossel), and Laurent Daudet (LightOn) for interesting discussions. RO acknowledges support by grants from Région Ile-de-France. MR acknowledges funding from the Defense Advanced Research Projects Agency (DARPA) under Agreement No. HR00111890042. FK acknowledges support by the French Agence Nationale de la Recherche under grant ANR17-CE23-0023-01 PAIL and ANR-19-P3IA-0001 PRAIRIE. Additional funding is acknowledged from "Chaire de recherche sur les modèles et sciences des données", Fondation CFM pour la Recherche-ENS.

## Footnotes

*Equal contribution. Corresponding authors {jonathan.dong, ruben.ohana}@ens.fr

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
