[Supplementary Material]

# A Convergence rate for translation-invariant kernels

**Theorem 2.** *(Rotation-invariant kernels) For the RI recurrent kernel defined in Eq. (9), under the assumptions detailed above, and with $\Lambda = 2\sigma_r^2 L$ (note the factor $2$ compared to Theorem 1). For all $t \in \mathbb{N}$, the following inequality is satisfied for any $\delta > 0$ with probability at least $1 - 2(t+1)\delta$:*

$$\left| \langle x^{(t+1)}, y^{(t+1)} \rangle - k_{t+1}(\Delta^{(t)}, ..., \Delta^{(0)}) \right| \leq \frac{1 - \Lambda^{t+1}}{1 - \Lambda} \Theta(N) \qquad \text{if} \quad \Lambda \neq 1 \qquad (20)$$

$$\leq (t+1)\Theta(N) \qquad \text{if} \quad \Lambda = 1 \qquad (21)$$

*with $\Theta(N) = \frac{4\kappa^2 \log \frac{1}{\delta}}{3N} + 2\kappa^2 \sqrt{\frac{2 \log \frac{1}{\delta}}{N}}$.*

*Proof.* Under the assumptions, Proposition 1 yields with probability greater than $1 - 2\delta$:

$$\left| \langle x^{(t+1)}, y^{(t+1)} \rangle - k([x^{(t)}, i^{(t)}], [y^{(t)}, j^{(t)}]) \right| \leq \frac{4\kappa^2 \log \frac{1}{\delta}}{3N} + 2\kappa^2 \sqrt{\frac{2 \log \frac{1}{\delta}}{N}} = \Theta(N) \qquad (22)$$

It means the larger the reservoir, the more Random Features $N$ we sample, and the more the inner product of reservoir states concentrates towards its expectation value, at a rate $O(1/\sqrt{N})$. We now apply this inequality recursively to complete the proof, based on the observation that both Eq. (11) and (12) are equivalent to:
$\left| \langle x^{(t+1)}, y^{(t+1)} \rangle - k_{t+1}(\Delta^{(t)}, ..., \Delta^{(0)}) \right| \leq (1 + \Lambda + \Lambda^2 + ... + \Lambda^t)\Theta(N)$.

For $t = 0$, provided $x^{(0)} = y^{(0)} = 0$, we have, according to Eq. 14, with probability at least $1 - 2\delta$:

$$\left| \langle x^{(1)}, y^{(1)} \rangle - k_1(\Delta^{(0)}) \right| \leq \Theta(N) \qquad (23)$$

For any time $t \in \mathbb{N}^*$, let us assume the following event $A_t$ is true with probability $\mathbb{P}(A_t) \geq 1 - 2t\delta$:

$$\left| \langle x^{(t)}, y^{(t)} \rangle - k_t(\Delta^{(t-1)}, ..., \Delta^{(0)}) \right| \leq (1 + ... + \Lambda^{t-1})\Theta(N) \qquad (24)$$

Using the Lipschitz-continuity of $k$, this inequality is equivalent to:

$$\left| k(2\sigma_r^2(1 - \langle x^{(t)}, y^{(t)} \rangle) + \Delta^{(t)}) - k(2\sigma_r^2(1 - k_t(\Delta^{(t-1)}, ..., \Delta^{(0)})) + \Delta^{(t)}) \right| \leq (\Lambda + ... + \Lambda^t)\Theta(N) \quad (25)$$

With Eq. (14), the following event $B_t$ is true with probability $\mathbb{P}(B_t) \geq 1 - 2\delta$:

$$\left| \langle x^{(t+1)}, y^{(t+1)} \rangle - k(2\sigma_r^2(1 - \langle x^{(t)}, y^{(t)} \rangle) + \Delta^{(t)}) \right| \leq \Theta(N) \qquad (26)$$

Summing Eq. (25) and (26), with the triangular inequality and a union bound, the following event $A_{t+1}$ is true with probability $\mathbb{P}(A_{t+1}) \geq \mathbb{P}(B_t \cap A_t) = \mathbb{P}(B_t) + \mathbb{P}(A_t) - \mathbb{P}(B_t \cup A_t) \geq 1 - 2\delta + 1 - 2t\delta - 1 = 1 - 2(t+1)\delta$:

$$\left| \langle x^{(t+1)}, y^{(t+1)} \rangle - k_{t+1}(\Delta^{(t)}, ..., \Delta^{(0)}) \right| \leq (1 + ... + \Lambda^t)\Theta(N) \qquad (27)$$

$\square$

# B Explicit Recurrent Kernel formulas

We have defined so far the general formulas of RI and TI Recurrent Kernels in Eq. (8) and (9). We will give now their explicit formulas for specific activation functions that one may encounter in Reservoir Computing.

Two reservoirs $x^{(t)}$ and $y^{(t)}$ are driven by two respective input time series $i^{(t)}$ and $j^{(t)}$. They obey Eq. (1) and in the infinite-size limit, their inner product converges towards an explicit Recurrent Kernel. In practice, one needs to compute the inner products for each pair of input time series, from the training or testing sets, that we concatenate to construct a kernel matrix.

A list of different activation functions and their associated kernels is provided in Table 3. Without recurrence, it is always possible to write the corresponding kernel as an integral that one may evaluate:

$$k(u, v) = \int dw \rho(w) f(\langle w, u \rangle) f(\langle w, v \rangle) \qquad (28)$$

where $\rho(w)$ is the distribution of the weights, usually an i.i.d. gaussian distribution. However, in all the cases presented here, $k(u, v)$ happens to contain inner products $\langle u, v \rangle$, which makes it possible to define the corresponding Recurrent Kernel.

| $f(\cdot)$ | Associated kernel $k(u,v)$ |
|---|---|
| Erf$(\cdot)$ | $\frac{2}{\pi}\arcsin\left(\frac{2\langle u,v\rangle}{\sqrt{(1+2\|u\|^2)(1+2\|v\|^2)}}\right)$ |
| RFFs: $[\cos(\cdot),\sin(\cdot)]$ | $\exp\left(-\frac{\|u-v\|^2}{2}\right)=\exp\left(-\frac{\|u\|^2+\|v\|^2-2\langle u,v\rangle}{2}\right)$ |
| Sign$(\cdot)$ | $\frac{2}{\pi}\arcsin\left(\frac{\langle u,v\rangle}{\|u\|\|v\|}\right)$ |
| Heaviside$(\cdot)$ | $\frac{1}{2}-\frac{1}{2\pi}\arccos\left(\frac{\langle u,v\rangle}{\|u\|\|v\|}\right)$ |
| ReLU$(\cdot)$ | $\frac{1}{2\pi}\left(\langle u,v\rangle\arccos(-\frac{\langle u,v\rangle}{\|u\|\|v\|})+\|u\|\|v\|\sqrt{1-\left(\frac{\langle u,v\rangle}{\|u\|\|v\|}\right)^2}\right)$ |

Table 3: Table of point-wise non-linearities $f$ and their approximated kernels. For any $u,v\in\mathbb{R}^p$ the kernel $k(u,v)$ is the limit when $N$ goes to infinity of $\frac{1}{N}\langle f(Wu),f(Wv)\rangle$ with $W\in\mathbb{R}^{N\times p}$ an i.i.d. normal random matrix. In the case of Reservoir Computing, we have $u=u^{(t)}=[\sigma_r x^{(t)},\sigma_i i^{(t)}]$ and $v=v^{(t)}=[\sigma_r y^{(t)},\sigma_i j^{(t)}]$. We observe that in this table, all kernel formulas depend only on $\langle u,v\rangle$, $\|u\|$, and $\|v\|$, which makes it possible to easily derive the Recurrent Kernel equations.

In our case, $u^{(t)}=[\sigma_r x^{(t)},\sigma_i i^{(t)}]$ and $v^{(t)}=[\sigma_r y^{(t)},\sigma_i j^{(t)}]$ so that:

$$\langle u^{(t)},v^{(t)}\rangle=\sigma_r^2\langle x^{(t)},y^{(t)}\rangle+\sigma_i^2\langle i^{(t)},j^{(t)}\rangle\rightarrow\sigma_r^2 k_t(l^{(t-1)},\dots,l^{(0)})+l^{(t)} \qquad (29)$$

when the reservoir size $N\to\infty$. Similarly, $\|u^{(t)}\|^2=\langle u^{(t)},u^{(t)}\rangle$ and $\|v^{(t)}\|^2$ are symmetric inner products that can similarly be expressed as in Eq. (29). Hence, the Recurrent Kernel formulas are derived from the previous one by noting that:

$$\lim_{N\to\infty}\langle x^{(t+1)},y^{(t+1)}\rangle=k_{t+1}(l^{(t)},\dots,l^{(0)})\equiv k(u^{(t)},v^{(t)}) \qquad (30)$$

Analytic formulas in more general cases may not exist and they would need to be replaced by successive integrals. In this work, we restricted ourselves to functions described in Table 1 with simple analytic formulas, to speed up the RK computation. For instance, the error function is very close but not equal to the hyperbolic tangent in our implementations of Reservoir Computing, and performance in practice is very similar.

The successive integrals can still be explicitly defined. Eq. (28) describes the asymptotic kernel limit for any arbitrary $(u,v)$. To define recurrent kernels, we need to express it as a function of $\langle u,v\rangle$, $\|u\|^2$, and $\|v\|^2$ only. This is possible thanks to the invariance by rotation of the gaussian distribution of $w$. Without loss of generality, we can thus assume that $u=\|u\|e_1$ and $v=\|v\|(\cos\theta e_1+\sin\theta e_2)$ with $e_1$ and $e_2$ the first two vectors of the canonical basis and $\theta=\langle u,v\rangle/(\|u\|\|v\|)$ (which is a function of the three quantities of interest). The multidimensional integral boils down to a two dimensional integral:

$$k(u,v)=\int\int dw_1 dw_2\rho(w_1)\rho(w_2)f(w_1\|u\|)f(\|v\|(w_1\cos\theta+w_2\sin\theta)) \qquad (31)$$

where $w_1$ and $w_2$ are gaussian random variables, projections of $w$ on $e_1$ and $e_2$. Hence it is possible to iterate Recurrent Kernels numerically, that are the large-size limit of any Reservoir Computing algorithm for every activation function $f$. Each component of the square kernel matrix would require the evaluation of this two-dimensional integral, it may be possible to use tabular values to speed up computation.

## C   Numerical study of the independence hypothesis

One assumption for the previous convergence theorems states the weight matrices $W_r$ and $W_i$ have to be redrawn at each iteration. This independence hypothesis is required in Eq. (18) and Eq. (26), to ensure that $x^{(t)}$ and $y^{(t)}$ are uncorrelated with the weight matrices. This is necessary in the theoretical study to properly define the expectations and ensure the i.i.d. requirement for the random variables in the Bernstein inequality.

However, this assumption is unrealistic for practical Reservoir Computing. Resampling weight matrices at each timestep is computationally demanding and output weights would depend on the realization of these random matrices: one would need to keep the same random matrices in memory for testing.

However, in Fig. 3, we investigate the convergence with and without redrawing weights at each iteration, and this independence hypothesis does not seem to be necessary: convergence is still achieved with fixed weight

Figure 3: Mean-Squared error between the kernel matrix obtained with RC/SRC with the asymptotic kernel limit, with and without resampling the random matrices at each iteration, to test the independence hypothesis of the theorem. $50 \times 50$ kernel matrices have been generated for all pairs of 50 random input time series of length 10. Several activation functions and their corresponding recurrent kernels are presented here. We observe that the hypothesis does not seem to be necessary since RC and SRC without resampling also converge to the RK limit at sensibly the same speed.

matrices. We show the Mean-Squared Error $\|K_1 - K_2\|_2^2/n^2$ between the kernel matrix $K_1$ from the explicit RK formula and $\hat{K}_2$ the one obtained with RC and SRC, with and without redrawing the random matrices at every timestep. Each kernel matrix is of size $50 \times 50$, as we use $n = 50$ random i.i.d gaussian input time series of dimension 50 and time length 10. Each curve is an average over 10 realizations and the reservoir scale is set to $\sigma_r^2 = 0.25$ to ensure stability.

We confirm the observation from Fig. 1 that the larger the reservoir dimension, the closer we are from the RK asymptotic limit. This is valid for several activation functions, the ones presented in Table 3. We also confirm that SRC generally converges faster than RC.

Convergence is still achieved when resampling the weights at each iteration, and speed of convergence is not significantly different than for the fixed random matrix case. Thus convergence seems to be much more robust in practice, and this may call for further theoretical studies.

# D  Stability of Reservoir Computing and Recurrent Kernels

As the reservoir is itself a dynamical system, it can be stable (differences in initial conditions vanish with time) or chaotic (differences in initial conditions explode exponentially). This is linked with the Echo-State Property, extensively studied for Reservoir Computing. It states that two reservoirs initialized differently need to converge to the same trajectory, provided they share the same weights (at each time step if weights are resampled). This property is important so that the reservoir state after a large enough time $\tau$ does not depend on the arbitrary reservoir initialization. Stability or chaos can be tuned depending on a set of hyperparameters. An important one is the scale of the reservoir weights: when small, initial differences get damped exponentially with time, whereas they may explode if reservoir weights are large.

We verify this Echo-State Property here for Reservoir Computing. In Fig. 4 we present the squared distance $\|x_1^{(t)} - x_2^{(t)}\|^2$ as a function of time $t$ between two randomly initialized reservoirs $x_1$ and $x_2$, for the same input time series from the Kuramoto-Sivashinsky dataset. A normalization factor has been added to normalize this distance to 1 at $t = 0$ and each curve is an average over 100 realizations. The activation is the error function, the input scale is set to a small value $\sigma_i^2 = 0.01$, and we vary the reservoir scale $\sigma_r^2$. For $\sigma_r^2 = 0.49$ and 1, dynamics are stable and the two reservoir states converge quite quickly to the same trajectory. When $\sigma_r^2 = 2.25$, dynamics becomes chaotic and the two reservoirs follow very different dynamics due to their different initial conditions.

Figure 4: Stability analysis of Reservoir Computing and Recurrent Kernels. We compute the normalized square distance between two reservoirs or recurrent kernels fed with the same input time series and different initializations. For RK or RC when $\sigma_r^2 \leq 1$ we see that trajectories converge to a single one after some time. This means that initial conditions are forgotten after a number of iterations. On the other hand, when $\sigma_r^2 = 2.25$ for Reservoir Computing, the reservoir is in a chaotic regime and always depend on initial conditions. It is interesting to observe that Recurrent Kernels are generally more stable than RC.

Recurrent Kernels may also present this transition from stability to chaos. Moreover, this stability property is important for Recurrent Kernels in practice. RKs need to be iterated a certain number of times, and thanks to stability this number of iterations can be reduced to the finite memory $\tau$ and not on the full length of the time series. This change reduces considerably the computational costs.

We thus also investigate numerically the stability of Recurrent Kernels, i.e. how they depend on the initial conditions. In Fig. 4, we present the normalized difference between two kernel matrices $\|K_1^{(t)} - K_2^{(t)}\|_2^2$ as a function of time, for two recurrent kernels $K_1$ and $K_2$ initialized with a matrix full of ones or of zeros, and fed with the same input time series, for the arcsine Recurrent Kernel corresponding to the erf activation function. We observe that Recurrent Kernels are in general a lot more stable than Reservoir Computing. This characteristic may be interesting to investigate further.

We may now draw an interesting parallel between this study and, as we unroll the Recurrent Neural Network through time, multilayer perceptrons with random weights, linked with compositional kernels. They correspond to our case, $i^{(t)} = 0$ for $t \geq 1$ and $i^{(0)} \in \mathbb{R}^d$ is the time-independent input. This stability property corresponds to a final layer that does not depend on $i^{(0)}$, and as such information does not flow in the deep network. Hence, whereas it is advantageous in Reservoir Computing to be stable, it may be detrimental for deep neural networks.

# E    Implementation details for Reservoir Computing

Several tweaks are useful to improve the performance of Reservoir Computing for time series prediction. We used the erf activation function as it is the closest from the hyperbolic tangent already used in Reservoir Computing, that still possess a simple Recurrent Kernel formula.

First, we add a random additive bias $b \in \mathbb{R}^N$ sampled from an i.i.d. normal distribution $\mathcal{N}(0, \sigma_b^2)$. The variance of this bias vector $\sigma_b^2$ is a hyperparameter to tune, like the variance of the reservoir or input weights. This bias helps to diversify the neuron activations in the reservoir. Hence, the reservoir update equation becomes:

$$x^{(t+1)} = \frac{1}{\sqrt{N}} f\left(W_r\, x^{(t)} + W_i\, i^{(t)} + b\right) \tag{32}$$

As stated previously, we concatenate the reservoir state with the last value of the time series we have received. Information about the past is still encoded in the reservoir, but with this simple change, the reservoir is rather used to compute perturbations on the current value, and does not have to reconstruct the whole spatial profile. We add a renormalization hyperparameter $r$ for this concatenation, in order to control the weight of the reservoir versus current input.

A hyperparameter search was performed, for a total of 5 hyperparameters (the reservoir scale, input scale, bias scale, the previous concatenation factor, regularization constant). Since there is a large number of hyperparameters to tune, we perform it on one hyperparameter at a time, going through the set of parameters several times. The final set of hyperparameters of Fig. 2 is $\{\sigma_i, \sigma_r, \sigma_b, r, \alpha\} = \{0.4, 0.9, 0.4, 1.1, 10^{-2}\}$.

Figure 5: How to use Recurrent Kernels for time series prediction. In Reservoir Computing, the input is continuously fed to the reservoir and all the reservoir states for every timestep $t$ are stored for training. With Recurrent Kernels, we construct $n$ small windows of the time series of length $\tau$ and compute scalar products between each pair using $\tau$ iteration of Eq. (8) or (9).

For completeness, we give here the exact definition of the Mean-Squared Error of Fig. 2. The target output $O(t) \in \mathbb{R}^d$ for $t = 1, \ldots, T_{\text{pred}}$ corresponds to the next states of the chaotic systems, and for each $t$, we evaluate the MSE between $O(t)$ and the prediction of the algorithm $\hat{O}(t)$, which is simply $\|O(t) - \hat{O}(t)\|^2 / d$.

# F   Implementation details for Recurrent Kernels

We also used a Recurrent Kernel to perform chaotic time series prediction. We chose an arcsine rotation-invariant kernel, the asymptotic limit of a reservoir with error function activations. We use the principle described in Section B, with the addition of a random gaussian bias that corresponds to adding a constant dimension to the vector $u^{(t)} = [\sigma_r x^{(t)}, \sigma_i i^{(t)}, \sigma_b]$.

Additionally, we have introduced for Reservoir Computing a concatenation step we need to reproduce with Recurrent Kernels. In RC, we concatenate the reservoir and the current input before computing the prediction. The corresponding operation for Recurrent Kernels is the addition of a linear kernel computed from all pairs of input points: $K^+_{kl} = \langle i^{(t)}_k, i^{(t)}_l \rangle$. This kernel matrix $K^+$ is added to the Recurrent Kernel after the iterations and before the linear model for prediction.

We also expand more on the process of generating the input data for Recurrent Kernels. In time series prediction, each reservoir state (neglecting a warm-up phase) is used during training to learn output weights to predict the future states of the system. Since there are $n$ training examples, this corresponds to an $n \times n$ kernel matrix. In the Recurrent Kernel setting, we train a linear model on the final kernel matrix. We thus construct $n$ time series of length $\tau = 50$ for each time step of the training data (neglecting the effect of edges), where the length $\tau$ is determined by the stability of the Recurrent Kernel. This process is depicted in Fig. 5.

# G   Recursive vs non-recursive prediction

Following previous strategies developed for chaotic time series prediction with Reservoir, RC, SRC, and RK algorithms were trained only to perform next-time-step prediction. To predict further in the future, this prediction is then fed back into the algorithm to iterate further in time. As explained previously, this defines an autonomous dynamical system that should be synchronized with the chaotic time series if training is successful.

Another possible strategy would be to use a given reservoir state to predict $T_{\text{pred}}$ time steps in the future. The output dimension $c = d\,T_{\text{pred}}$ is larger and the learning task becomes more difficult.

We show here the usefulness of this strategy based on autonomous dynamics. In Fig. 6, we show the performance of Reservoir Computing prediction on the Kuramoto-Sivashinsky dataset, with and without recursive prediction. With recursive prediction (left), this corresponds to the strategy already presented in Fig. 2, and it is not surprising that prediction up to at least 2 Lyapunov exponents is possible. Without recursive prediction (right), the algorithm has a much harder time to predict the future of the chaotic system. Instead, after a short while, it only returns the average value of the time series.

Note that the same hyperparameters were used in both cases. While it may be possible to improve the performance of the direct prediction strategy, by increasing the size of the reservoir or playing with regularization parameter, but we show here the simplicity and effectiveness of the recursive prediction strategy.

Figure 6: Comparison of recursive and non-recursive prediction. We see that with recursive prediction (left), Reservoir Computing is able to predict quite precisely up to at least 2 characteristic times. On the other hand, without recursive prediction, Reservoir Computing quickly has a hard time to guess the future of the KS system and outputs its mean for long prediction times.