[Reviews · NeurIPS 2020]

Review 1

Summary and Contributions: This paper establishes a link between Reservoir Computing (RC) and Recurrent Kernels (RK) by showing that Kernel Methods (KM) with Random Features (RF) can be equivalent to RC and thus RC can converge to a deterministic kernel as the number of neurons N tends to infinity. The link is clarified through RK as a limit of such approximation. The authors also present Structured Reservoir Computing (SRC) as an alternative to RC (and thus RK) that is more memory efficient and yields faster prediction times. Results are supported by empirical studies on random and chaotic time series.

Strengths: The paper is well written, and the grounds for claims and the links between the different elements are well established in Section 2. All claims are supported by corresponding empirical results or complexity analysis.

Weaknesses: As an uninitiated person to the almost all components studied here, it is hard for me to detect flaws in the theoretical derivations shown here, which are at the heart of the paper. I list some clarity aspects below on the parts I could follow, but I could not detect any major flaw.

Correctness: The theoretical claims are proven and empirically supported.

Clarity: Overall the paper is very well written and even though I could not follow all the mathematical derivations, the ideas were clearly presented, the text is easy to follow and the supporting evidence is presented clearly. From Equation (6) and downwards, there are a lot of occurrences of expressions in the form of k(.) for which k(.) is applied on a single component. All previous uses of k(.) define the kernel operation applied on two components but it is unclear to me what this operation is when applied on a single component.

Relation to Prior Work: The links with previous works seem clearly defined and put in the relation to the current work. Note that I am unfamiliar with the prior work.

Reproducibility: Yes

Additional Feedback: I wish to apologize to the authors for such a poorly constructive review on the hard work presented here. I do not think I have the required expertise and theoretical background to further comment on the work. ======================== Given the rebuttal and other reviewers appreciation I keep my positive score. My lack of expertise in the area prevents me to further comment.


Review 2

Summary and Contributions: This work aims to explore the kernel limit for Reservoir Computing. The authors proved and tested its convergence theoretically and empirically. The authors further proposed methods, namely recurrent kernel and structured RC, that showed the proposed methods are more memory and computationally efficient than conventional RC. ----------update----------- Thanks for the detailed response, my concerns were addressed. I'd like to increase my score

Strengths: Training RNNs remains challenging, RC could be one of the approaches to simplify this and yield more efficient RNN models.

Weaknesses: 1. The applicability to other real-world tasks is unknown 2. Some experimental settings/results are unclear, see my comments in "Additional feedback"

Correctness: 1. I am not able to judge the correctness of convergence theorem 2. The main claim is not well supported by current empirical results

Clarity: Could be improved, particularly experiment section could be clear

Relation to Prior Work: Relation to prior work is well presented

Reproducibility: Yes

Additional Feedback: 1. In Table 2, The results are averaged on how many different runs? For measuring the time costs, each different run would lead to quite different results, a single run number is not reliable. Why does KR have the same number for all N? . 2. Is the conclusion section missing? 3. In line 181, it says the convergence can be achieved even though the assumptions in previous theoretical study are not satisfied, then what is the point of the theoretical study? 4. I am a little bit confused by "forward" and "train", usually train stage of RNNs contains "forward" and "backward", here the "train" means the part of linear regression only?


Review 3

Summary and Contributions: The authors connect two important methods, Reservoir Computing (RC) and kernel methods, both theoretically and with experiments. By comparing Recurrent Kernels (RK) to structured and unstructured RC neural networks, they show that both methods converge numerically to the same limit. They attempt to derive this limit theoretically but with very restrictive assumptions compared to practice. They also show that their structured RC and RK methods can reduce computational complexity and are able to predict a chaotic time-series accurately.

Strengths: Understanding how neural networks behave in the RC setting, where the network is initialized with random weights and an output is trained with linear regression, can help us understand the inner workings of Recurrent Neural Networks (RNNs) in general, how to better train them in machine learning and how they behave in models of the brain in neuroscience. Establishing a link between kernel methods, random features and RC, is an important and novel contribution in helping the community study RNNs further. The paper was well written and all theoretical claims and proofs were well-stated and explained. Demonstrating that RC nets converge to their kernel limit is an important finding, yet some of the assumptions were too restrictive. Numerical experiments further demonstrated the authors claims and they were well explained and plotted.

Weaknesses: One weakness of the paper is that there is somewhat of a gap between theory and practice. The necessary components for their theory to hold is to have random network and input weights for every time-step, while keeping the network stable and exhibiting the echo-state property. While the latter might be possible by selecting certain hyperparameters, having random weights is never done in practice or in their experiments.

Correctness: My two main concerns in this paper are regarding the theoretical assumptions necessary for RC convergence to its kernel limit: 1 - Random Weights: Their theory assumes that the weight matrices of the network (W_r) and input (W_i) are randomly redrawn every time-step. The authors explain that this assumption was necessary as correlations are difficult to take into account yet no further explanation was given. The main problem with this assumption is that it does not reflect the use of RC networks in practice where its weights are fixed. Ideally there should be a theoretical exploration of fixed weights and if that it is too difficult at least an explanation on why both random and fixed weights give similar convergence as demonstrated in the Supplementary Figure 3. 2. Stability Analysis: One of the theoretical assumptions of the paper is that the function f is bounded by a constant K. That means that the network must be stable and also exhibit the echo state property. The authors explain that by having a finite memory this property is achieved and they demonstrate stability for the dataset as input in Supplemental Figure 4. What is not clear to me is in the theoretical setting of random weights, how are we guaranteed stability and/or the echo-state property, that are necessary for convergence?

Clarity: The paper is well written and all theoretical claims and numerical results were well presented.

Relation to Prior Work: Relating kernel and random feature methods to reservoir computing is a novel approach in this paper, while discussed previously by [5], the theoretical and experimental findings of this paper is a new contribution to the field. The authors clearly discuss relations to other contributions in the introduction. I suggest considering discussing Inubushi & Yoshimura 2017, especially concerning network memory and the echo-state property.

Reproducibility: Yes

Additional Feedback:


Review 4

Summary and Contributions: The paper introduces structured reservoir computing as an efficient approach to RC and prediction of chaotic time series, together with recurrent random kernels. The paper formally connects and compares RC and recurrent kernels with structured reservoir computing, including a comparison of computational complexities / memory complexities for the different approaches.

Strengths: The work establishes formal connections between RC and random features. Convergence is investigated and along with complexity analysis is a major and important component of the paper. The claims appear to be sound and the overall work is relevant to the NeurIPS community. I have not previously seen these connections being investigated, and believe the work would be interesting for implementations on resource-constrained hardware. The work clearly points out limitations of the different approaches.

Weaknesses: The paper makes quite good use of the available space: while it would be interesting to see additional empirical evaluation in the main part of the paper, there is nothing in the paper that should be replaced. The focus on only chaotic time series prediction makes sense in light of the original ESN papers, even though it would be nice to see additional applications.

Correctness: As far as I can tell claims and methods are correct. The paper contains only limited empirical evaluation (but I believe that no additional empirical evaluation is necessary).

Clarity: The paper is very well organised and well written.

Relation to Prior Work: Important relevant prior works are discussed and cited, and the paper manages to place itself well into their context.

Reproducibility: Yes

Additional Feedback: It should be possible to reproduce the major results; however reproducing the practical aspects of the work may be difficult.

[Author Response · NeurIPS 2020]

1. We would like to thank the reviewers (R1, R2, R3, R4) for the very constructive reviews on our work, pointing out
2. the merits and raising interesting questions to answer. We received very positively the comments on the quality of the
3. presentation and we believe this constructive discussion will greatly improve the quality of the manuscript.

4. In this work, we propose a link between kernel methods and fixed random weights Recurrent Neural Networks (RNNs),
5. quoting R3, "**an important and novel contribution in helping the community study RNNs further**". We also
6. leverage this theoretical correspondence to accelerate Reservoir Computing using structured transforms, from $O(N^2)$
7. complexity to $O(N \log N)$ for the main reservoir computation of Eq. (1). Even though there is a gap between theory
8. and practice, we argue these ideas bring significant computational savings, calling for future theoretical studies. We
9. would like now to answer the reviewers' comments thoroughly.

10. @R2, @R3: **Assumptions of the theoretical study.** Our Theorem requires the assumption of resampling the weights
11. matrices at each time, which is never done in practice. This assumption remains important to obtain a sum of i.i.d.
12. random variables to apply Proposition 1. Nonetheless, actual implementations match this theory well and the conclusions
13. of the theorem still bring interesting insights. For example, the convergence rate of the Mean Square Error matches
14. the $1/N$ scaling of the theory, and we show in Fig. 3 of the appendix there is clearly no difference with and without
15. redrawing the weights.

16. We recognize this theoretical study is limited and we have tried to present its limits clearly and honestly in the manuscript.
17. To complement it, we show with direct numerical computations and practical applications the convergence of RC
18. towards its Recurrent Kernel limit. To discuss R2's comment about the reason why we propose a theoretical study, we
19. believe it remains essential to explain rigorously the behavior of our ML algorithms, even if conditions sometimes have
20. to be relaxed to obtain informative results, as this paves the way for future studies.

21. @R1, @R2, @R3: **Broader impact.** To discuss the broader impact of the presented work, we will add to the manuscript:
22. (1) theoretical studies to understand machine learning (ML) are important to avoid relying on black boxes, as more and
23. more applications appear in our daily life; (2) efficient ML is necessary due to the ever-increasing power consumption
24. required for computation.

25. We deeply think this work is establishing a connection between random RNNs and kernel methods that will open
26. up future studies on this important topic in machine learning. This is the reason we have submitted this work to a
27. conference such as NeurIPS. We will now proceed with the answers to the more technical questions:

28. @R1: **Kernel function of 1 or 2 variables.** $k(x, y)$ is indeed a function of two variables, and we use the simplifying
29. notation for translation-invariant kernels $k(x - y) \equiv k(x, y)$ and rotation-invariant kernels $k(\langle x, y \rangle) \equiv k(x, y)$. This
30. precision will be added to the manuscript.

31. @R2: **"How many runs for the time benchmark in Table 2?"** We did not have to average this timing benchmark as
32. the standard deviation of this measurement is negligible on a GPU (less than 1% at $N = 1,000$), both for the matrix
33. multiplication and inversion for each network dimension.

34. @R2: **"Why does RK [Recurrent Kernels] have the same number for all N?"** RK corresponds to the limit of RC
35. when $N$ tends to infinity, and as such, does not depend on $N$.

36. @R2: **Clarification on "forward" and "train" steps:** Since internal weights are not trained in RC, we first compute
37. the network states with Eq. (1) (this is the "forward" step), and training is performed separately with linear regression
38. (no "backward" step is necessary).

39. @R2: **Absence of conclusion.** We have chosen to summarize our results in the "Main contributions" section of the
40. introduction for clarity. However, we will add a conclusion to discuss future lines of work in the next version.

41. @R2, @R4: **Additional applications.** We have chosen to focus on chaotic time series prediction, a promising yet
42. challenging application for RC. This choice has been motivated by the substantial amount of prior works and the
43. particular interest shown recently by the RC community as it is well said by R4 "*The focus on only chaotic time series
44. prediction makes sense in light of the original ESN papers, even though it would be nice to see additional applications.*"
45. While additional applications would surely be interesting, it is beyond the scope of this paper to find novel applications
46. of RC, for space reasons (as R4 pointed out, "*The paper makes quite good use of the available space* [...] *there is
47. nothing in the paper that should be replaced*").

48. @R3: **Stability and Echo-State Property for resampled random weights.** Stability and the ESP can also be
49. described with resampled weights: one can check whether two reservoirs initialized differently converge or not to the
50. same trajectory, provided they share the same weights even with resampling at each time. We would like to thank the
51. reviewer for this very relevant remark and for the reference that will be added to the manuscript. We believe stability is
52. an essential question to investigate further for RK, as this property is central in Reservoir Computing.

[Meta-Review · NeurIPS 2020]

Reviewers unanimously liked the idea and novel connections between Reservoir Computing (RC) and Recurrent Kernels (RK) and the link between Kernel Methods (KM) with Random Features (RF) to be equivalent to RC. There were few minor concerns that were aptly addressed in the rebuttal. Post rebuttal discussion made it very clear that this is one of the best papers in many reviewers lot! We hope authors will use the feedback to make a better camera ready version.